# Examining the Implementation of the Performance-Based Financing Equity Strategy in Improving Access and Utilization of Maternal Health Services in Cameroon: A Qualitative Study

**DOI:** 10.3390/ijerph192114132

**Published:** 2022-10-29

**Authors:** Miriam Nkangu, Julian Little, Constantine Asahngwa, Raywat Deonandan, Roland Pongou, Orvill Adams, Sanni Yaya

**Affiliations:** 1School of Epidemiology and Public Health, University of Ottawa, Ottawa, ON K1N 6N5, Canada; 2Health Promotion Alliance Cameroon (HPAC), Yaounde 14315, Cameroon; 3Denis and Lenora Foretia Foundation Cameroon, Yaounde 14315, Cameroon; 4Interdisciplinary School of Health Sciences, University of Ottawa, Ottawa, ON K1N 6N5, Canada; 5Department of Economics, University of Ottawa, Ottawa, ON K1N 6N5, Canada; 6School of International Development and Global Studies, University of Ottawa, Ottawa, ON K1N 6N5, Canada; 7The George Institute for Global Health, Imperial College London, London SW7 2BX, UK

**Keywords:** performance-based financing, equity, maternal health, grounded theory, poor and vulnerable

## Abstract

Performance-based financing (PBF)—a supply-side strategy that incentivizes health providers based on predefined quality and quantity criteria—introduced an innovative approach to reaching the poor by means of using PBF equity instruments. These PBF equity instruments include paying providers more to reach out to poor women, selecting services used by the poor, subsidizing user fees to reduce out-of-pocket expenses, and adding complementary demand-side intervention. Before the implementation of the PBF equity instrument in Cameroon, there were few initiatives/schemes to enable the poor to access maternal health services. Moreover, there is a significant research gap on how the equity elements are defined and implemented across contexts. This study aims to understand (i) how health facilities define and classify the poor and vulnerable in the context of PBF, (ii) how the equity elements are implemented at the community and facility levels, and (iii) the potential impact on access to and the use of maternal health services at the facility level and challenges in the implementation process. We used key informant interviews and focus group discussions (FGDs) based on a grounded theory approach to gain an understanding of the social processes and experiences. Data were collected from three districts in the Southwest region of Cameroon from April 2021 to August 2021. Data were transcribed and analyzed using MaxQDA. The thematic analysis approach/technique was used to analyze data. Key informant interviews and focus groups were conducted with 79 participants, including 28 health professionals and service administrators, 27 pregnant women, and 24 community health workers in three districts. Health facilities employed various subjective approaches to assess and define poor and vulnerable (PAV) persons. Home visits were reported to have an impact in reaching the poor and vulnerable to improve access to maternal services. Meanwhile, a delay in the payment of PBF incentives was reported to be the main challenge that had a negative relationship with the consistent provision of care to the poor and vulnerable, especially in private health facilities. The theory generated from our findings suggests that the impact of the PBF equity elements specific to maternal health depends on (i) a shared understanding of the definition of PAV among different stakeholders, including providers and users, as well as how the PAV is operationalized (structure), and (ii) the appropriate and timely payment of incentives to health facilities and health providers.

## 1. Introduction

Performance-based financing (PBF)—a supply-side strategy that incentivizes health providers based on predefined quality and quantity criteria—introduced an innovative approach to reaching the poor, known as the PBF equity instruments [1,2,3]. These PBF equity instruments include the following elements: (i) paying providers more to reach out to poor women; (ii) selecting services used by the poor; (iii) subsidizing user fees to reduce out-of-pocket expenses; (iv) adding complementary demand interventions; and (v) using community health workers to address information and social barriers [1,2,3] (Table 1 below). The intention is to facilitate the allocation of more resources for people who have more needs: thus, a health facility in a rural or poor area would be allocated more resources than those in urban or more affluent areas. More resources are allocated to health facilities where people are poor, and a health facility can reduce the cost of services but still earn more money from PBF equity (incentives) to cover the cost of services.

In Cameroon, based on information gathered from the PBF offices, for some curative services where health facilities require out-of-pocket payments, there is an indicator to provide free or subsidized services to those considered poor. These equity elements aim to ensure that health facilities reach out to the poor and provide services; however, regarding identifying and selecting poor people, health facilities and communities are left to decide how to identify those who should benefit from the services under the category of poor and vulnerable [1,2]. To implement this component, health facilities and the community can employ the following criteria, as proposed by Fritsche et al.: (i) proxy means testing—a method by which each household is scored based on one or more of a small number of easily observable characteristics or assets; (ii) community targeting, where village committees or groups within the village decide who should be considered poor within a health catchment area; and (iii) identifying people circumstantially vulnerable as a result of road accidents or internally displaced persons [1,2].

In Cameroon, again based on information gathered from the PBF office, these equity elements are operated at the regional, health facility, and individual levels. Regional equity incentives aim at reducing poverty differences between regions. For example, the northern regions are considered the most vulnerable, so they are allocated a higher equity incentive. Within the regions, there are also district-level differences, and the distinctions regarding equity incentives are based on the district’s characteristics and distance to the district hospital. At the individual level, people are classified into two groups according to their poverty level: as poor (that is, extremely poor) or not poor. 

According to the information gathered from the PBF team, PBF contracts are signed directly with health facilities based on a predefined equity expectation. Each health facility is assigned an equity category, based on accessibility, population density, poverty level, and distance from the district health facility. The equity category is ranked from 1 to 5. Category 1 indicates that access is difficult in all seasons and that the population is very sparse, very poor, and far from the district health hospital. Category 5 indicates that the health facility is highly accessible in all seasons, serves a higher populated area, and has a population that is closer to the district hospital. This equity score is used to calculate the equity incentives for each health facility based on predefined quantity criteria for each indicator.

At the individual level, an equity bonus is provided per service rendered within the list of indicators considered under the category poor and vulnerable (PAV), and the incentives vary as a function of the economic class in which the patient is classified. For example, PBF pays a greater incentive (4×) for a skilled birth delivery by qualified staff if the woman is defined as PAV than if she is classified as non-poor. This aims to ensure that the health facility will not create financial inaccessibility for the person. This applies to most of the indicators, including vaccination, normal deliveries, antenatal care visits, minor surgeries, major surgeries, and C-sections.

Ultimately, the above-listed equity elements aim to improve access to maternal and child health care and to stimulate demand by providing incentives to health providers, to reduce out-of-pocket expenses and information barriers to expand coverage for the PAV, and thus to increase access to and the use of maternal health services [1,2]. Therefore, this study focused on the health facility and individual levels because these equity elements are operationalized at the facility level, and the trickle-down effect is expected to impact individuals considered PAV. It should be noted that PBF is implemented at all health facilities, be it primary, secondary, or tertiary care facility; however, in the context of this study, there are only primary and secondary care facilities within the study region. The study examined the above equity components, with a focus on antenatal care (ANC), skilled birth delivery, and family planning.

Previous studies have reported a lack of evidence that PBF influences equity [1,4,5]. In Rwanda, PBF implementation indicates a pro-rich policy but did not include an equity-based strategy to increase the use of ANC and skilled birth delivery [6,7]. However, in Tanzania, PBF implementation was found to be a pro-poor policy without an equity component [8]. Findings from our unpublished systematic review on the PBF-induced effect on out-of-pocket expenses to improve access to antenatal care and skilled birth delivery indicates that some studies used data from demographic and health surveys and quantified poorness using the household characteristics and asset index [9]. This may differ from the way PBF defines and categorizes being poor [9]. Other studies used impact evaluation data (most of which did not assess the PBF effect on equity) and quantified poorness using the household characteristics and asset index [9]. However, one study from Burkina Faso reported this as a potential limitation in that the criteria used in defining poverty in the PBF differed from the criteria used in the study [9,10]. Our systematic review findings point to the existing call for research in understanding how the PBF equity instruments are defined and implemented in various contexts [1,4,5]. Our systematic review also reported that studies did not define poor according to the PBF criteria, which may potentially lead to missed reporting or missed classification, thereby putting the equity dimension of this initiative to question [9]. This implies, from a broader perspective, that the concept of poor and vulnerable has not been well defined, not only within the PBF equity dimension but also in the context of equity in maternal health services. 

Based on the current literature, there is a significant research gap regarding how the PBF equity instrument is defined and implemented in various settings and contexts [4]. Given that Cameroon adopted PBF as one of the national health financing strategies towards achieving universal health coverage, it is important to investigate how PBF equity elements are defined and implemented. Moreover, the PBF equity elements is a new strategy for reaching the poor in a health system such as that in Cameroon, where few or no services target the poor to improve access to maternal health services. This study aimed to close this research gap by exploring how the PBF equity elements are defined and implemented at the health facility and individual levels in Cameroon. 

## 2. Methods

### 2.1. Theoretical Framework and Study Design

The study is a qualitative exploratory cross-sectional study that employs a grounded theory approach to gain an understanding of the social processes and experiences regarding (i) how health facilities define and classify PAV; (ii) how the PAV classification is implemented at the community and facility levels; (iii) the challenges for the health facility during the implementation process; and (iv) the potential impact on access to and the use of maternal health services at the facility. Grounded theory focuses on social meaning and action and is usually employed in qualitative methods to study aspects of social and individual interactions, especially when a phenomenon has not been well studied [11,12,13,14]. It enables the exploration of experiences from multiple perspectives to inform the data [11,12,13,14]. The goal was to identify the key factors and actors in implementing the PBF equity strategies, to categorize the relationships of the elements, and to develop a theory on the process that influences the outcome of the PBF equity elements in the context of maternal health services in Cameroon.

This study employed Charmaz’s constructivist grounded theory approach, which presumes that researchers are co-producers of knowledge and not mere observers of the social processes [15,16]. It assumes that theories and meaning derived from data are socially constructed, influenced also by the researcher’s interaction with the data as well as with their study participants [15,16]. However, we continue to use the principal that our interpretation remains grounded in the data, taking into full account the context in which such data were collected [15,16].

The first part of the study assessed the equity elements using an equity lens—PROGRESS [17] (Appendix A)—to describe how equity was considered in the design and how it was applied in the context, and this is presented in Appendix A. The second part of the study examined the definition of PAV and how it is identified and categorized within the context, and the implementation processes, challenges, and gaps, and presents a theoretical framework grounded from participants’ own words. The third part of the study combined the research findings to inform the PBF equity strategies in the context of antenatal care and skilled birth delivery in Cameroon.

### 2.2. Context/Settings

Cameroon is one of the six countries that make up the Central African Region, with a population estimate of 26 million inhabitants in 2020 [18]. It has a decentralized health system with a pyramidal structure comprising operational (district levels), regional, and central levels. Cameroon finances health care mainly through out-of-pocket (OOP) expenditure [18]. Women comprise approximately 50% of the population, and life expectancy at birth was 59 years in 2021 [18,19,20,21]. Health is delivered in both the private and public sectors, and PBF is implemented in all sectors. The private sector comprises for-profit and confessional health facilities. Confessional health facilities are also called faith-based facilities or mission health facilities (these faiths include Catholicism, Presbyterianism, and Baptism). This study was conducted in three districts (Buea, Limbe, and Tiko) in the southwest region of Cameroon, in which there has been ongoing armed conflict since 2017.

### 2.3. Sampling

Sampling techniques in grounded theory can begin with individuals in the study area who are informed and could offer a rich source of data [20]. Grounded theory involves purposive and theoretical sampling in selecting study participants [15,16,22]. Determining the sample size in qualitative studies is not straightforward [15,16,22,23,24,25,26,27]. Authors have proposed different sample size guidance [15,16,22,23,24,25,26,27]. For grounded theory, Creswell proposed a sample size of 20–30 participants [26], while Morse proposed 30–50 participants [27], with no justification for the above-mentioned numbers [22,23,24,25]. However, other factors influence the final sample size, including the study objectives, study design, population characteristics, data analysis approach, and availability of resources, such as budget and time [12,13,23,24]. 

Purposive sampling was used to identify key informants for the interviews, such as health providers and PBF administrators, and subsequent participants were identified using snowballing techniques. This study assessed data saturation, which included documenting the steps in developing the themes, determining the number of themes generated by several interviews, and identifying the code frequency and hierarchy [13,23].

All the health facilities in each district were eligible for the interview until saturation was reached. For each district, health facilities were grouped according to sector. The number of skilled birth deliveries in 2020 was used to rank health facilities. The first five most used health facilities for skilled birth delivery and the five least used health facilities were sampled for key informant interviews across sectors. This process continued until saturation was reached. The sampling frame ensured facilities with equity categories of 5 and 1 were included.

Theoretical sampling in this study was based on data that were obtained during concurrent interviews and was used to identify participants to be invited for further interviews and focus group discussions (FGDs). We sampled a group of women considered as PAV for FGDs. Based on categories generated from the interviews with the PBF administrators, we sampled another group of women considered non-PAV (NPAV) for an FGD and later sampled PBF verifiers for interviews. During FGDs with community health workers (CHW) and the interviews with clinical verifiers and health facility administrators, two other actors involved in the process were identified, namely “head of quarter” (appointed local leader for a specific health area) and community verifiers. This led to interviews with quarter heads and community verifiers. In addition to the interview process, a PBF validation meeting was scheduled during this period, and this was intended to gain additional insight on the challenges faced by health facilities in the verification and validation process of PAV. Subsequently, we accompanied a PBF verifier to the field to observe the verification and validation of the indicators related to the PAV. Finally, we interviewed district regulators as a follow-up based on data gathered during the interviews with PBF coordinators.

### 2.4. Data Collection Techniques and Instruments

Interviews were conducted with health facility administrators, PBF administrators, health care providers, and district medical officers. FGDs were conducted with pregnant women and community health workers. An administrator for each eligible health facility responsible for assessing the PAV was sampled for interview. The initial samples were interviews with health providers and PBF stakeholders, and focus group discussions with community health workers and women considered as PAV. In order to gain a comparison of the definition of poor and vulnerable by the various actors identified from the initial codes and from the experiences in the implementation process of the equity elements, the decision to conduct FGDs with non-PAV and interviews with district regulators, quarter heads, and the verifiers alongside observation of the verification process was made as part of the theoretical sampling. The use of non-participant observation further explored the emerging categories to understand and capture some aspects of the social processes and the interaction amongst health providers and the PBF verifiers and to gain further insights into participants’ responses.

Providers and administrators (including the PBF team) were interviewed at their offices. For CHWs, pregnant women, and community verifiers, a preferred location and time were arranged at their convenience. Considering that theoretical sampling is driven by concepts or categories [15,16,24], we did not have a fixed number of participants to sample. This is consistent with grounded theory principles, assuming that saturation is reached when participants are interviewed at different stages as the theme emerges; this is compared with subsequent data collection until there is no new information regarding the data being collected. The interview guide was pilot tested with six potential informants not selected for the study. They discussed the questions with the investigator orally, providing feedback on readability, applicability, and understanding. Minor feedback-related changes were made by the investigator to ensure that the participants understood the subject matter and that the questions could capture the data of interest to the study.

Open-ended questions [15,16] were used for interviews to enable a broader discussion, especially as PBF is a newly implemented reform and little has been documented about the PBF equity strategy. Participants were prompted as needed to elicit additional information. All interviews with health providers were conducted at the health facility and started with a warmup question to allow a comfortable environment for the participant, followed by general and specific questions and, finally, the conclusion [23]. Warmup questions for providers started with, for example, “Tell me about your health facility and your role in the health facility.” For women, they included, “Tell me about your last pregnancy and where you sought care.” Intermediate questions [15,16] for providers included, “Could you please tell me about your experiences with your health facility under the PBF?” and for women, “Can you tell me your experiences with consultation fees for antenatal clinic services?” These were followed by probing regarding specific, ANC, skilled birth delivery, and related costs, followed by questions on quality of care for the same services and probing to compare with previous pregnancies. Specific questions were more focused, especially following questions such as, “Can you explain to me what you mean by or what you think about cost, care, for example, for ANC with a previous baby and a recent baby?” For providers, probing focused on the equity strategies for the PAV; specific questions for providers related to how they defined and categorized the PAV, including challenges and facilitators; and exploring specific questions concerning the context. Closing questions [16] summarized and allowed participants to add information not covered during the interview process [23], such as “Is there anything that you would like to add or recommend to the PBF or suggest changes you would like to see in the implementation process?” The closing question helped the interviewer explore the participants’ experiences in more depth [16]. Questions for PBF managers resembled those for providers, with minimal adjustment depending on the individual role. However, at the regulatory level, discussions were informal, with specific questions concerning quality control and facility accreditation, such as the facility not meeting quality standards; discussing strategies in place; or addressing the challenges discussed, possible regulatory measures, and the potential for sustainability.

Field notes were collected throughout the data collection process to record relevant nonverbal considerations or actions, such as the tone, mood, and coherence of respondents. Interviews lasted 30–45 min. No relationship was established between the interviewer and participant before the start of the interview, and the study purpose was explained to the participants before the interview, followed by informed consent, which was read to participants. FGDs were conducted with community health workers at a convenient location, and each FGD lasted for 1 h. Interviews and FGDs were conducted face-to-face with only the interviewer and the participant. The data were transcribed with the help of research assistants. Demographic data were collected on age, marital status, and educational attainment. All interviews were recorded, and participants were informed and provided consent before recording. Interviews were conducted using MaxApp (v1.1.3, VERBI Software, Berlin, Germany) [28]. Interviews and FGDs were conducted by MN using the local dialect, “pidgin English,” except those with health providers and managers, which were conducted in English. The lead investigator (M.N.) is familiar with the local spoken language, so no translator was required. Interviews were conducted from April to August 2021, and member checking and validation were conducted in July 2022.

### 2.5. Data Analysis

Data analysis in grounded theory proceeds simultaneously with data collection [15,16]. In this light, the constant comparative method was used as the researcher compared data during the collection process. This helps produce a conceptual understanding to better explain the processes, experiences, and understanding of the PBF equity components and the implementation approaches and challenges [16]. The interviews were transcribed and exported into the MaxQDA software (v2020, VERBI, Germany) [28]. In line with Charmaz’s approach for data analysis, we used three stages of coding—initial, focused, and theoretical [16]. The initial codes were developed using free codes and the in vivo coding within MaxQDA [28], which allows line-by-line coding using the participants’ own words. Focused coding involved grouping the initial codes into categories and conceptualizing the process. This process involves constant comparison and the use of memos in categorizing the groups. The notes were grouped into a coding scheme to create subcategories. These subcategories were compared, contrasted, and revised by S.Y. and J.L., and later subcategories with similar content, trends, and understanding were merged to produce main categories. Relevant or dominant codes were selected, and categories were generated by grouping the remaining codes around the dominant codes. Dominant codes were identified by considering the number of times or frequencies of specific codes, the number of times they were applied to the data, and the types of participants linked to the code. Finally, we narrowed all the main categories and developed core categories; theoretical coding was then used to represent participants’ experiences in implementing the PBF equity elements [16].

Two independent coders, M.N. and C.A. independently coded the data using various themes to assess the inter-rater reliability and trustworthiness of the study. Inter-coder reliability was discussed between the two coders for different and similar interpretations to enrich and fine-tune the analyses and to ultimately converge them into a shared interpretation of the data. Saturation for grounded theory was reached by returning to the participants to compare the data collected until the new data indicated no new information.

Comparisons of the data within and between transcripts identified potential relationships between categories and subcategories. For example, data were triangulated across different stakeholders interviewed to compare and capture experiences and to understand the interaction between actors in the process of identifying and providing services to the PAV in implementing the PBF equity strategies. In this theoretical sensitivity process, developed categories and relationships were iteratively refined following the conclusions of FGDs and analysis of FGD transcripts. The main categories were refined, and the relationships between main categories using insights from theoretical memos were used to expand and compile additional information about the data to help explain the results. 

### 2.6. Trustworthiness

The trustworthiness of qualitative research is important to ensure that the study is rigorous enough to produce findings capable of impacting policy or practice [29,30]. For confirmability, the decisions made in the research process, from the research objectives to the interpretation of findings, are thoroughly described [29]. Feedback on the research objectives and questions and on the interpretations of the findings was received from the co-authors. This was vital for establishing dependability and credibility in the findings, as they provided valuable assessments of factual errors and competing interpretations in the manuscript. This study employed various methods of data triangulation as it involved the collection of data from women; providers; health facility administrators in public, private and confessional health facilities; PBF administrators; and regulators. These multiple data sources helped produce greater depths and breadths of understanding of the study findings [29,30]. The study also employed method triangulation using key informant interviews, non-participant observation, and FGDs to collect data. Member checking was also employed from June to July 2022 by asking selected key informants based on the breadth and depth of the data to assess the data interpretation and to reflect on what they expressed to the researcher. There was no major change to the data interpretation. This study has met the following quality conditions for grounded theory: iterative process, theoretical sampling, theoretical sensitivity, coding and memoing, constant comparisons, and theoretical saturation. Constant comparisons and theoretical saturation helped ensure that the results represented and explained the data [29,30].

### 2.7. Ethics

The Bruyère Research Institute at the University of Ottawa approved the study (# M16-18-057), and administrative clearance was obtained from the Faculty of Science, University of Ottawa (#H-02-19-2829). In Cameroon, we obtained ethical approval from the University of Buea Faculty of Health Sciences (Ref 2020/1342-02/UB/SG/IRB/FHS) and administrative clearance from the regional delegation of public health in the southwest region (Ref R11 MINSANTE/SWR/RDPH/PS864/715). We obtained informed consent for the survey from all participants who agreed to participate.

## 3. Results

Participants were interviewed across 25 health facilities in the three districts (6 for-profit, 3 confessional, 2 para-public, and 14 public). Key informant interviews and focus groups were conducted with 79 participants, including 28 health professionals and service administrators, 27 pregnant women, and 24 community health workers (see Table 2a,b). The age range for women was 21–41. Equity scores for the health facilities ranged from 20 to 30. With the ongoing crisis, the definition of PAV extended to internally displaced people, and the equity score percentage was increased. There were approximately 39 maternal health indicators, most of which had options for PAV. 

Characteristics of respondents from health districts are presented in Table 2.

### 3.1. Applying an Equity Lens in Defining the Poor and Vulnerable in the Context of PBF Equity Strategy

The use of the PROGRESS (place, religion, occupation, gender and sex, race, education, social and economic, social capital) plus tool has been widely recommended as a reminder of policy on equity considerations [17]. Applying the PROGRESS lens [17] to the PBF equity design elements indicates that some of the variables were considered in the design phase (Appendix A). We observed that both individual and contextual factors are considered in the PBF equity design elements. However, in the implementation phase, health facilities selectively applied some of the items as applicable to their context and convenience. This means that, depending on the state of the individual upon presentation at the health facility, they have an equal probability of being categorized as PAV irrespective of their actual household income quintile, occupation, or education. These criteria are subjective as there are no specific means for an objective assessment; this therefore depends on the discretion of the health care providers and their relationship with the patient. Appendix A describes the actors involved and application of each PROGRESS [17] item within the context.

#### Themes Generated from Participants Responses

The main themes generated in the study are presented in Table 3 and constituted the relationship in the framework illustrated in Figure 1. The relationship between dominant categories reported in the implementation process of the PBF equity elements and their definition and perspective of poor and vulnerable constituted the framework in understanding the implementation process of the PBF equity elements and potential impact on PAV. For example, delay in payment was a dominant category that emerged across all sectors and stakeholders and had a negative relationship with the provision of care to the PAV. This relationship provides meaningful understanding to the theory that has been developed, and it is the original contribution of knowledge in this research. Thus, the theory generated from this study suggest that the impact of the PBF equity elements specific to maternal health depends on (i) shared understanding of the definition of PAV among different stakeholders, including providers and users, as well as how the PAV is operationalized (structure), and (ii) the appropriate and timely payment of incentives to health facilities and health providers.

The results are presented as follows: the first part provides a description and interpretation of how PAV is defined based on the themes grounded from the data, and the processes involved in the identification and validation of PAV and the actors involved. The second part reports the challenges in the process and describes the relationship from the themes that constitute the theoretical framework and support the theory developed from this research.

### 3.2. Definition and Classification of PAV by Health Facilities

Each facility presented a set of criteria for defining PAV. These criteria included occupation; marital status; inadequate finance to pay for services at the point of care; internally displaced status; employment status; disability (mainly physical); disease exposure status (mainly HIV status for women); place of residence (quarter); and poor family background, defined as the level of social network and lack of job or employment status. These factors are also similar to the characteristics of the PROGRESS framework [17] (Appendix A). Table 4 below presents the reported criteria used by health facilities to identify and categorize PAV across sectors. The health facilities reported different ways of defining, identifying, and categorizing who was PAV within their community and at the point of care. However, some variables were common to all the facilities, such as internally displaced people.

Most participants reported aspects related to occupation, employment status, and internal displacement. The household characteristics were used to explore observable items such as owning a television and living conditions, and this relates to the proxy means testing. This was conducted more by community health workers as they were more engaged in the exploration stage for potential PAV individuals and referral to the health facility. Vulnerability was mainly defined based on disease status; internally displaced people, given the ongoing conflict in this region; widowed status; and physical disability (Table 5). Health administrators, general supervisors, and social service workers explained how they define and identify PAV:


*“Before now we identify the poor and vulnerable from the orphanages but with the current war we have a lot of internally displaced persons”*

*(Health administrator, private facility Buea)*



*“We use various approaches to identify poor and vulnerable such as conduct psychosocial assessment to know if they can afford to pay the bills because some people will want to pretend to be poor”*

*(Social worker, public facility)*



*“Those who are unable to work, physically disabled or are sick are considered vulnerable because vulnerability is circumstantial but to be poor can change” *

*(Provider, Public Limbe)*


Given that this study was conducted in a conflict setting, most internally displaced people who used the health facilities for specific services could be classified as PAV at the health facility. However, in the context of PBF, a circumstantially vulnerable person could also be PAV, for example, due to an accident leaving the person vulnerable, irrespective of their income status. In such a situation, the individual is treated in the health facility as PAV. All health facilities are expected to classify the PAV in the register for specific maternal health services separately from the NPAV. This classification is further verified and validated by the PBF verification and validation team before the assigned PBF incentives can be attributed to the indicator. 

### 3.3. Actors in the Identification and Referral Process of PAV

The process is implemented in two ways, through referral from CHW, as assigned by health facilities, or through the social service department, as explained in health facilities that have social service workers. However, this is not limited as all patients can be considered a potential PAV at the point of care depending on how the patient presents their situation upon making payments.

Health facilities work in collaboration with the CHWs and quarter heads because they know most of those who live in the community. With PBF, CHWs are becoming more active in some health facilities. CHWs are given a referral slip, which allows them to refer patients to the health facility, and each successful referral is an incentive. Health facilities conduct a quarterly meeting with the CHW, and they are instructed that if they identify that someone meets the conditions for PAV during the referral procedure and referral process, they must indicate this in the referral slip and ask the quarter head to endorse it. The GS explains:


*“We usually have meetings with these community health workers every three months, and we brief them on who they are supposed to refer to the hospitals as PAV…during the briefing, we tell the community workers to also find out why that person is a pauper, because if they are not, they can lose the incentives”*

*(GS, Limbe)*


### 3.4. Steps in the Identification of PAV

Of the three approaches listed above to identify poor people from the PBF manual [1], the two most used approaches in this region were community targeting and identifying circumstantially vulnerable people.

### 3.5. Community-Level Targeting

Health facilities are expected to recruit community health workers in their health areas according to the health facility’s needs. Health facilities sign contracts with CHWs for referral and follow up during home visits, which are sometimes conducted together with nurses or during community random spot checks. Once a CHW identifies a potential PAV individual in the community, the CHW is expected to validate this with the quarter head, who then endorses their signature, and the CHW refers the PAV person to the health facility for treatment. At the health facility, another validation process is conducted by the general supervisor (GS) or other administrator assigned to this task. At this phase, the GS or assigned administrator conducts another assessment of the patient to ensure they meet the criteria mostly based on the factors listed in Table 4. Depending on the assessment outcome, the GS decides whether the patient will be considered as PAV and whether they qualify for free or subsidized services.

### 3.6. Health Facility

Besides referrals from a CHW, health facilities also identify PAV as circumstantially vulnerable people—this may include road accidents, internally displaced people, and those with physical disabilities. This is at the discretion of the provider or health facility administrator based on the presentation of the patient at the point of care. It could also result from circumstantial factors such as being internally displaced; an accident, which does not necessarily relate to the level of income or employment; and other types of assessments conducted, as listed above. Additionally, during consultation at the point of care, health providers may refer a patient to the GS for validation as PAV based on a combination of the factors listed above.

### 3.7. Social Service Workers

Regional social welfare services were run by the government before PBF to support poor patients who could not pay their bills; this was at a minimal scale and only at the regional health facilities. With the introduction of PAV, health facilities that initially had a designated person for this service use the same service to assess PAV individuals and then refer the patient to the GS for approval. Individuals from other government health facilities can be referred to the regional health facilities to benefit from this service; however, this was reported to be more theoretical than practical, given the limited resources. Nevertheless, with the introduction of PBF, all health facilities are obliged to provide services for PAV with the assumption that this is a context that has more PAV and there is no way a facility would say that they do not have PAV individuals. This has helped to increase the number of patients benefitting from the services, as reported by the social service department.

The social service team does not work with the CHWs. They only identify PAV people at the level of the health facility upon care, and they can follow up at home after treatment to ensure that they adhere to treatment and provide psychosocial support when needed, as two of them stated:


*“We have our techniques and observation from the patient presentation and talking and some try to tell lies so we go further even to their homes to verify if they have a family member who can support their bills and we educate some on family responsibility.”*

*(Social worker, Public include town)*



*“With PBF the difference is that PBF has made us to be smart in identifying some of the PAV because before we had challenges with the administrator in identifying but with PBF we go further and we have the strength to explore and even go to the facility for example before if we have 5 paupers we will present to the administrator and they will only support like 2 but with PBF when we present 5 the administrator will cover all the five because PBF will pay part of the bill.”*

*(Social service worker, Public)*


### 3.8. Actors in the Verification and Validation Process for PAV for the PBF Equity Bonus

During the verification process, the PBF verifier explained to us that since the inception of PBF in these districts, the PAV list was designated to quarter heads, but the PBF team realized that quarter heads were listing the names of their families and household members. This was changed, and PBF transferred the assignment to the health facilities for management committees to identify and validate the PAV. Each health facility has a management committee that provides a focal point for the community and contains a CHW and the GS of the facility. The chief of the center or any board management team member must endorse the PAV cases treated at each health facility for PBF verification. Any case not signed by the administrator (chief of center or committee board member) is not considered PAV or cannot be validated, which may result in an error margin and eventual cancellation of the indicator if the error margin exceeds 10%.

All facilities are expected to declare a PAV person for each indicator; otherwise, an explanation must be provided, or they may face sanctions, given that most people cannot pay bills in Cameroon. It is not practical to report no PAV. Therefore, health facilities are compelled to report PAV cases for specific indicators.

The home visit register gives a reliable picture of the quarter and PAV; for each family visited, all the names of the family or household are listed. All home visits and quantities declared must be validated and justified before PBF buys, for example, if a home visit is on family planning and the registers indicate that more family planning services were bought and aligns with the timing of the home visits or campaigns for community awareness. However, if vice versa, home visits are related to something else or registers do not align with the issue, further justification is required for validation.

### 3.9. Challenges in the Identification and Validation of PAV

#### 3.9.1. Health Providers’ Challenges

Some GSs reported that some people pretend to be PAV, and during the assessment process, they later realize that those people can pay their bills. Some providers reported challenges with how CHWs identify the PAV and that some CHWs are not well-trained and assume everyone referred to the health facility is automatically considered PAV or will be given the services under that category. According to the provider, there are criteria that must be evaluated at the health facility at the point of care before confirming a PAV individual, and they also have quotas. An individual is not always given care under this category even if they are referred by a CHW. Referral by a CHW to a health facility under PAV does not necessarily imply that the individual will be considered PAV at the point of care by the facility. This poor understanding or communication between the facility and CHW was reported by both parties and sometimes caused friction.

#### 3.9.2. Community Health Workers’ Challenges

Some CHWs reported challenges when referring individuals, where these individuals did not meet the administrators on arriving at the health facility and became frustrated when they were not given the services. However, not all those referred by CHWs are considered PAV at the point of care. Some CHWs reported challenges in the referral; when they referred potential PAV candidates to health facilities, they were not given the services, and this miscommunication on both sides sometimes resulted in conflict, as one of them stated:


*“Management gives instructions to inform pregnant woman that haven’t started antenatal (ANCs) to come for consultations, promising to offer them reduction in their bills, but these promises are not respected. The bills are the same, I haven’t seen any differences in the reduction of amount with respect to the bills for the poor, because when you bring those woman and what they expect, it is not the same set up as earlier promised, they start blaming me why I didn’t inform them before hand to prepare themselves before coming, instead telling them something else and coming here, they are telling them something else….*

*(FGD;CHW Tiko)*


Community health workers also reported challenges with quarter heads. CHWs reported times when the quarter head could not identify or know all the people living in that area; thus, regarding the PAV, when CHWs notice someone is sick, they approach the quarter head, explain the situation to him, and plead with him to identify whether the person can afford the bills before taking them to the health center. If the quarter head cannot do this, he must ask his assistant to help him perform that task. One of the CHWs shared their experience:


*“Yes in my own quarter when you go to see the quarter head, the quarter head will tell us to go, because the health center doesn’t see them as anything” *

*(FGD;CHW Buea)*


Interestingly, there were conflicting discussions with quarter heads. Some were unaware of the PAV referral process in their quarter and did not understand how the CHWs operated. Conversely, some quarter heads had different opinions, and some had not heard about PBF and the CHW referral system for the PAV, as one of the quarter heads stated: 


*“I have heard about CHW as they mention them sometimes during the meetings at the health facility, but I have not seen or worked with PBF or have community health workers come to me. The CHW don’t come to me and we will like to work with them so we can identify paupers in the community”*

*(quarter head Buea)*



*“I think PBF publicity aspect is very dormant, they haven’t really mobilized people in a way that they could be aware that there’s some help available. They could do it on the radio, but not everyone really listens to the radio. So I don’t know if they could come to the community, meet up the various quarter heads so they could work together so as to show they are really out to provide help. Because as for now it’s just a few of us who even know about PBF, so this shows their publicity is really slow”. *

*(Quarter head Buea)*


#### 3.9.3. PBF Manager’s Challenge and Delay in Payment

From the perspective of PBF managers, it was apparent that services were provided to people (considered PAV) once they were in the health facility, even if they did not have the finances to pay; however, this was not well-implemented or managed in certain health facilities, as expected. We further explored this aspect amongst providers, and a key relationship was identified between the delay in PBF equity bonus payment and the delivery of services to the PAV, which could impact access and the use of maternal services. 

Concerns were expressed regarding poor management of the PAV by most facilities, as they did not meet the 10% threshold for equity services. This implied potential underuse of the services by the PAV, as expected, due to implementation challenges for some health facilities. Issues were reported regarding some facilities not meeting their maximum 10% threshold for PAV, yet these facilities had enormous unpaid bills by women who had delivered and could not explain why the unpaid bills were not treated as PAV given that the facility had not reached its maximum 10% threshold.

Questions were asked to determine what caused health facilities’ inability to meet the threshold to manage the PAV effectively. These were answered by some health facilities.


*“The thing is that their payments are not regular, you are expected to present a business plan, provide services to poor and vulnerable and pay staff, when the payment do not come on time you start asking if you are there to pay staff or to take care of the poor and vulnerable”*

*(Provider, private facility Buea, Tiko, Limbe)*


### 3.10. The Role of PBF Equity Strategy in Improving Access and Utilization of Antenatal Care and Skilled Birth Delivery

Despite the challenges reported by health care providers in the identification and validation process of PAV, some women considered as PAV expressed positive experiences with access to services, especially not being asked for payment before treatment, which helped reduce barriers to financial inaccessibility for them. Some expressed their experiences with the support they received from the health facility after delivery, which was not the case before PBF.

### 3.11. Home Visits

The use of home visits as introduced in the PAV program was reported as an important mechanism in reaching out to the PAV by all the actors involved. This was not a practice before PBF, as reported by health facilities. Providers reported an observed influx of women in their clinics or health facilities due to the introduction of home visits. Providers saw PBF home visits as a mechanism helping to increase access to and the use of services by the poor by breaking information and social barriers through using CHWs for home visits. During a home visit, a health facility team is accompanied by a CHW. They educate and sensitize households regarding specific health-related conditions, including women who avoid using the health services due to cost and prefer to stay at home. This approach, reported by providers, helps identify women who would otherwise be unable to access the services due to cost as reported by a health provider.


*“With PBF we have the integrated outreach home visits that we do every month we have 18 communities, and we target each per month …this activity is very good because with PBF there is money that you can pay the staff and CHW and the staff are willing to go for home visits and we are also evaluated by that services and beside the activity also create an impact in the community because some community does not know our facility and also an opportunity for us to advertise our activities when we go like that they get to know that public facilities are also cheaper and we also see the reality of the population and we see the state and feel sympathy for some…”*

*(Provider, Limbe and Buea)*



*“We get to see the reality during home visits, I remember one time what we saw there I really felt pity we met three children from the same mother with serious wounds on very risky positions on their body and we felt so bad and we saw them being expose to potential infection and we had to invite the family to the health facility and treated them and follow up on their wounds until they got treated and this is because of the PBF equity support. Now the woman comes to the health facility for treatment…..Without PBF it will not have been possible at all because who will go out to the community without any motivation, no one will do that, the outreach and the home visits would not have been possible, the staff will not go out without motivation”*

*(Provider, Limbe and Buea)*


Community health workers also reported the importance of home visits but had a contrary view of how they were treated during them. They believed that without them, the nurses would be unable to conduct the home visits effectively as required by the PBF indicator because it is the CHWs who understand the community and the quarters, and providers acknowledged that without the CHWs, they would be unable to effectively implement the home visit. However, some CHWs expressed dissatisfaction regarding how they were paid for the home visits.


*“Last year home visit we were not being paid so we were paid only when we refer people and after that when they call I just told them I can’t be taking a nurse for a home visit she is the only one that was being paid and me taking her to the houses is not being paid so I refuse working for the home visit but with this new contract they have agreed to pay me and I have decided to go for the home visit”*

*(FGD; CHW, Tiko)*


The non-poor and vulnerable had a different perspective; they saw the cost of services as extremely high and expected a reduction from PBF. Meanwhile, one private for-profit facility reported that they have instituted some policies due to the PBF equity strategy to support transportation for women, feeding, and delivery of supplies.

## 4. Theoretical Framework in Understanding the Implementation and the Challenges of the PBF Equity Strategies at Health Facility and Community Level

### 4.1. Design Phase

Definition of PAV: The definition and identification of PAV at the point of care are highly subjective. This study found two separate concepts being used: “poor” and “vulnerable.” These had the same meaning but were defined and measured differently. PAV is defined in PBF as extreme poverty, which applies to providing services to those considered within the lowest income quintile. It is noteworthy that “poor” could mean anything to anyone at the point of care at the health facility. Alternatively, the definition of “vulnerable” in the context of PBF is circumstantial; it could relate to how the patient presents themselves at the point of care (accident, lack of money, and internally displaced people) but does not necessarily mean someone is poor based on the extreme poverty consistent with the scale used to categorize income quintile.

### 4.2. Implementation Phase

(i) Contextual Definition of PAV: PAV is mainly defined at the level of the facility, and it is at the discretion of the health provider or health administrator to categorize who is PAV to be eligible for subsidies or free services. This could introduce a potential problem regarding whether people are classified correctly to reflect extreme poverty related to the lowest income quintile. These are potential implementation issues, and health facilities are applying this classification differently based on their expectations and in different situations and scenarios.

Therefore, the PBF rationale for a 10% threshold in providing services for the PAV is not only based on targeting the lowest income quintile, given that most of the criteria used at the health facility relate to vulnerability. This vulnerability is a concern because, according to PBF, situations exist where even a senior divisional officer or regional delegate may be considered as vulnerable at the point of care, perhaps due to circumstantial factors, but this does not necessarily mean that they fall within the lowest income quintile. Therefore, circumstantial criteria relate to the ease of accessing services at the point of care without creating any financial inaccessibility, but this does not mean that it has an impact or it makes a difference in the poorest income group. It may instead increase the use of services amongst the richest quintiles. This may also explain some of the findings in other countries were PBF was seen as a pro-rich policy [6,7]. The advantage of vulnerability is that it can be more objective than subjective because it depends more on the circumstances in which the patient presents themselves; however, this does not imply that they are poor based on the lowest income quintile.

(ii) Objective Identification of PAV: the limitation of using the facility to identify the poor relates to the fact that household income or expense-related data that usually quantifies the household income quintile are not collected at the facility level. Therefore, data obtained from approaches such as interviewing or conducting social inquiries at the facility to identify who is poor at the facility level contradict the household survey data used to categorize the wealth index for poor incomes; this would instead apply to vulnerability. Secondly, the poor can be perceived and defined differently by individuals and their community, and this may be applied differently. Facilities mainly used the circumstantial approach and each facility applied different approaches. This is at the discretion of health facilities to identify the poor as most of them were internally displaced, and before PBF, the focus was more on orphanages, as some facilities reported. Thus, continuing identification of the poor using the facility at the discretion of providers may result in missed classification and even fraud. These factors should be considered, and those considered in classifying households into low-income quintiles for the use of maternal services should be considered in identifying the PAV since they consider various changes over time and space.

(iii) Type of Health Facility: Considering that Cameroon delivers care through the public and private sectors and the private sector comprises both confessional health facilities and private for-profit, the private sector provides up to 50% of the population’s care. This is important in designing and implementing all policies and interventions. Some confessional and private for-profit health facilities not meeting the threshold value for PAV requires additional enforcement strategies. There appears to be an organizational culture of not paying more to health workers, especially in confessional facilities; however, this is also based on payment delays. Facilities that rely on the PBF equity incentives to recruit and retain staff or to use the incentive to invest on other aspects are unable to retain staff or to plough back profit due to delays in payment. Although participants embraced the innovative approach of the PBF equity package, challenges remain since the provision of PAV services strongly depends on the health facility. For example, based on previous findings, the cost of antenatal care and skilled birth delivery increased with the introduction of PBF quality indicators in some health facilities [31]. The PBF equity strategies may be viewed as an alternative mechanism to partially offset this increase in cost by reducing out-of-pocket expenses for the poor, who are more responsive to changes in cost [8]. Given the increase in cost specifically for the first ANC visit and skilled birth delivery in some health facilities, objective identification and classification of the PAV are required to produce potential changes in the use of ANC and skilled birth delivery by the poorest quintile.

(iv) Verification/validation process Providers reported some challenges in relation to the peer-review verification process. They reported that reviewers have different judgment and interpretations of the indicators, and they are not sure if it is bias or misunderstanding. Moreover, some indicators need to be revised because some of the indicators were listed in line with some of the old practices in the previous system and those practices have been abolished with PBF. They believed there is a need to discuss the quality checklist together with them to agree and make recommendations to ensure both parties are on the same page. 


*“The issues with the indicators can even be serious than I am able to present. I know the verificators on the field are trained but somehow a clinician judges something differently from just a verificator on the field who is working strictly on those indicators and there are areas we really do not agree on. Clinically, you have reasons, but because they have to work as per their indicators it becomes difficult. For example, PBF has a definition for sexually transmitted infections. They want you to show that you had a positive chlamydia test, or a positive syphilis test but a clinician has the right to make the clinical diagnoses even if there is no laboratory confirmation. If a man has ureteral discharge, it’s an STI until proven otherwise; even if he cannot prove from a laboratory point of view to confirm that it’s an STI. In that case it counted but the criteria are not met, it is an over-reporting for which you will be penalised. And we lose money from the penalty”*

*(Provider, Buea and Limbe)*


(v) Structural and Managerial Issues: all actors are involved in identifying and validating the PAV. This makes the process more cumbersome, and the actors misunderstand the criteria for eligibility. However, some CHWs reported that they were asked to refer patients to the health facility and to indicate when they saw a need for the PAV classification. CHWs must also validate this process with the quarter head; however, some quarter heads are not well-versed with the community and households, and some require motivation. Health facility staff reported that some CHWs do not understand that not all referrals imply validation or approval; they are coached to ensure they do not make promises to the patient upon referral, and this causes tension between the health facility and CHWs, as the CHWs report losing their income. Additionally, health facilities also lose income during PAV validation, which could result from errors in identification and documentation.

Some facilities reported that the cost to treat a pauper is not equivalent to what is received in return to offset this cost. Some proposed an increase in the pay package and number of paupers considered because if the health facility continues to pay more than what PBF reimburses for pauper cases, it may affect the finances of the health facility in the long term, and this may not help its support of the PAV.

Issues were raised in some health facilities regarding readmitting a pauper and making them repeat the same procedure. Some PAV individuals are followed up with their social network to educate them on family responsibility, and changes were reported, as much negligence was attributed to ignorance. Therefore, some facilities reported that they continued to educate families on how to take care of loved ones when they were alive rather than focusing on a culture of taking care of their deaths.

(vi) Delay in Payment and Impact on PAV: this was a common theme across all levels and sectors and has been documented in the literature in other countries. This was reported to be the main challenge, and it was related to the expectation of providing services to the PAV, especially in the private sector. For example, providers reported that PBF has empowered them to be able to hire staff; that they have to pay the staff from PBF incentives; that it becomes challenging to manage such staff with delays in payment; and sometimes, that there is transfer aggression on the patient and this can potentially lead to old practices that have been abolished with the introduction of PBF, as reported by a health provider:


*“Before PBF we had private practice where the staff get something from what patients pay out of pocket but with the coming of PBF, it is no more because PBF took up the motivation of staff. So now we can have private practice when PBF does not meet up payment. So, the delayed payment is one of our major challenges”*

*(Provider, Tiko)*


### 4.3. Reporting Stage

(a)Poor documentation of poor and vulnerable

The PBF team reported poor registration of data due to a lack of collaboration among staff and poor supervision. Initially, there was a focal point responsible for entering data; however, this was not feasible since it failed if one person could not work, and it was challenging to find a replacement who understands how to enter the data. The data were verified before the verifier arrived. It is noteworthy that some facilities failed to manage the PAV well and that they abandoned files because of an inability for patients to pay bills. Some health facilities would not include the names from the abandoned files as PAV even when the facility was unable to meet the threshold for PAV because they hoped the families will eventually pay at some point. 

(b)Meeting the Threshold of 10%

Some facilities are not meeting the 10% threshold required for the PAV; this was attributed mainly to payment delays and was mainly common in private facilities. Management continues to provide implementation strategies to encourage each health facility to create a pauper committee in their respective areas with opinion leaders; religious leaders; and even representatives from the centers, such as the chief of the center from this committee. This committee can decide based on their context who is considered poor or define poor based on their context. They can easily identify the poor because, in each community, they know who cannot afford basic needs. Instituting paupers’ committees in all the health areas, where lists of the poor are compiled, will improve the management of the PAV; however, this must be considered alongside constant payment of the PBF equity bonus to achieve smooth implementation. Developing a paupers’ community is essential alongside using CHWs to identify new and eligible members who can be referred to join the committee based on specific criteria considered fair and agreed within each context.

(c)Quality of care and impact on Facility Type

Participants have long appreciated the quality of care in confessional and private facilities that encouraged them to use these services despite the high cost compared to public facilities. Quality of care, specifically regarding provider communication strategy, is important because the influx of patients in confessional facilities is likely to affect the categorization and the volume of PAV benefitting from subsidized services. More patients in such health facilities could increase the income to support the PAV. This may also mean that more patients cannot pay their bills or may pay them gradually based on findings from our previous studies. In addition, women reported that they preferred to attend the confessional health facilities because they understood their payment mechanism. Sometimes, they must remain and work to pay their bills. Due to the harsh treatment (attitude of providers) they receive in public facilities, they prefer to attend confessional facilities. Finally, the influx of patients may also affect the 10% threshold for these health facilities. The quality standard must be reinforced across public facilities. 

(d)Impact on access and utilization of antenatal care and skilled birth delivery

The PBF design elements for equity may not apply in all contexts in terms of implementation as designed, due to various contextual factors. In Cameroon, the heterogenous nature of the health system can influence the way some of these elements are implemented within and across sectors. It is likely that the PBF design elements did not consider some of these heterogenous characteristics despite the impact of an evaluation including health facilities across sector; however, the analysis and reporting did not consider such potential differences that may exist across sectors [21]. For example, our earlier study found that, as PBF attempts to improve the quality of care, the cost for the first ANC visit increased in public and confessional facilities. This increase also has an impact on access as it affects women’s attitude in shopping around for affordable services during ANC

Most of the health facilities reported similar characteristics in defining poor and vulnerable; there were still important differences in the way each sector would consider categorizing poor and vulnerable, and this had an impact on access and utilization amongst the poor. For example, private facilities reported that this was like the first time they were providing services to the poor for free or subsidized because they are for profit, and some of the facilities have dropped down their prices with the aim to motivate and stimulate demand but the prices are still high compared to the other sectors, while confessional facilities would provide options for those who cannot pay their bills to pay in instalment over time.

In addition to the limitations discussed above in terms of PBF definitions and identifying and classifying the PAV, the PBF equity strategy also has limitations regarding the provision of services because it is not expected that all the PAV will be provided services at the point of care given its subjective nature. Secondly, there is a cap of 10%, so if a facility is receiving an influx of PAV patients, this may skew the results in one direction, especially as participants prefer to attend ANC in public services, for example, due to cost, and to obtain services in private facilities during delivery due to quality [31]. Private facilities are hesitant to provide services under the category of PAV for several reasons relating to payment delays. Table 6 below indicates findings triangulated from this research and earlier study described elsewhere [31,32].

## 5. Discussion

The key findings of this study indicate that introducing a PBF equity strategy has positively impacted the overall motivation of health providers (doctors, nurses, and midwives) in delivering care to those considered poor. It is also considered a mechanism to provide services to the PAV at the point of care. This study revealed experiences and perceptions from both the demand and supply sides. From a supply-side perspective, health care providers expressed positive experiences and saw PBF equity instruments as a novel approach to reaching the poor but with challenges concerning payment delays. These payment delays have been reported in the literature [8,10,33]. From the demand side, some women reported changes in providers’ attitudes concerning care quality, which corroborates the findings from an earlier study from Cameroon [34] and out-of-pocket expenses for specific maternal services, and some experienced the provision of free services for the first time in health facilities or a payment reduction, especially for C-sections. These reported changes by some women in relation to cost reflect some of the findings in Tanzania in the way PBF facilitated access to skilled birth delivery [33].

Health facilities are expected to meet a threshold value of 10% per indicator for PAV, but most facilities do not meet this expectation, especially confessional facilities. Health facilities with social service departments mainly use the social welfare unit and interview the patient to evaluate whether they meet the criteria; this is only conducted at the point of care. If a patient cannot pay their health bills, they are referred to the social welfare or the assigned administrator at the health facility, who determines whether the patient meets the criteria listed in Table 2. This is solely at the discretion of the health facility or the assigned individual.

The irregularity of the PBF payment mechanism shapes how providers engage in sustaining aspects of the PBF equity strategy, and the motivation and quality of services are likely to decrease if the payment mechanism is not addressed. Objective identification of the PAV is critical for proper classification, and meeting the 10% threshold by health facilities is important in increasing the utilization of services by the poor. The PBF equity strategy provides for the PAV; however, implementation challenges in identifying and classifying the PAV can affect the outcome.

### 5.1. Policy Implications

The World Bank proposes three options to identify people who are poor and vulnerable within the PBF equity program [1]; however, it is evident that there is a lack of theoretical consensus in the definition of poor and vulnerable, and this can be problematic as “poor” can be defined and interpreted differently from one context to another. This may apply not only in the context of the PBF equity elements but also in a broader perspective on how policies in place are defined to identify and categorize who is poor and vulnerable within a specific context to ensure appropriate identification and classification of individuals in addressing equity issues in maternal health services. 

Quality of care is a strong indicator, drawing many women to private facilities, such as confessional facilities, which cannot meet the 10% threshold due to several factors such as payment delays. This affects the services they would have provided to the PAV due to limited finance to run the facility, fear of the unknown, risk of providing free services to the PAV, and precautionary measures these facilities are engaged in. Thus, improving quality in public settings with strict enforcement strategies will help balance some of these aspects. Appropriate enforcement strategies should be implemented to ensure that facilities meet the minimum quality standard. There should be policies in place to reduce the delay in payment. A demand-side mechanism such as home visits is also an encouraging mechanism, but it is necessary to properly educate and train the community health workers on careful identification and the expectations of the equity program. This structure can help enhance health promotion activities for the community and health facility. The social service team can work with the CHW more closely to enhance the identification process of the PAV. Research could further explore the impact of PBF equity program on the perceived health of individuals and inequality in accessing care.

### 5.2. Research Implications

This paper proposed a theoretical framework and established the need for further research and directions regarding the data to be collected in assessing PBF equity elements in the context of maternal health services. It assumes that assessing the effect of PBF equity elements on maternal health services requires an understanding of how the PAV is defined, the implementation process, and the challenges given the complexity of the concept. First, the use of grounded theory provides insight and explanations regarding providers’ experiences and the processes, and meaning they attribute to this policy. The findings also provide inputs or information for other countries implementing PBF equity strategies. Future research should assess how the management committees for PAV are structured and function and how their performance is evaluated.

Methodological limitations may arise in assessing equity if data are only focused on demographic and health survey household data. This is because the definition of PAV and the identification and validation at the facility level posed significant challenges that could result in misclassification bias. It is necessary to develop an adequate methodology to assess equity in PBF, considering the instruments, definition, and implementation of the equity elements. The World Bank PBF tool kit proposed relevant methodological approaches to measure inequality [1], but it is likely that this approach to using household characteristics and asset index may not provide an accurate assessment in the context of the PBF equity program without considering the definition of PAV and the identification, implementation, and validation processes. The use of CHWs only helps provide referrals or identify potentially vulnerable individuals who may not necessarily fall within the lowest income quintile and does not guarantee that they will be given care as a PAV patient upon reaching the health facility. Therefore, based on our findings and our theoretical framework, we have proposed the following approach to data collection in Table 7 to be considered when assessing the effect of PBF on equity on maternal health services.

### 5.3. Study Strengths and Limitations

Given the gap in the literature on how the equity elements are defined and implemented in various contexts, few studies have focused on examining the differential effect of PBF on equity [7,8,10]. This study is one of the first to examine how the PBF equity elements are defined and implemented in a specific context. The findings suggest the need to understand how these elements are defined in order to guide data collection and the scale employed in assessing the effect of PBF on equity. The framework can also help direct data collection for the effective study of equity and its various stages and indicate what to consider regarding data collection and potential bias. The framework can also be used to assess the PAV package to ensure it meets the context standard and not necessarily the facility perspective. It can also help monitor the future design of studies related to equity in defining PAV and methodological challenges regarding assessment since collecting data using a scale to quantify income when the contextual definition and classification of PAV are not based on the same scale or income classification is an important methodological error, and the results may likely not reflect the situation. Additionally, there may be some important differences in the definition of PAV in the context of the study given the ongoing conflict that included internally displaced persons within the category of PAV. This definition may vary from regions that are not affected by conflict.

Limitations concerning saturation in the grounded theory approach arise because only the person analyzing the data can confirm that saturation has been reached [15], so it was necessary to ensure all questions were answered. However, the use of triangulation and member checking is a strength in minimizing the bias and providing a broader perspective and validation in the interpretation of the data. This study is limited to the health facilities in the Southwest region, which may differ from the French-speaking region; however, it provides an overview of the processes and challenges that may be common across sectors in all regions given the national scale of PBF.

## 6. Conclusions

No single path exists for achieving universal health coverage (UHC), but all journeys toward achieving UHC start with bold commitments toward equity and quality of care. Equity is actualized in UHC, with the goal of ensuring that all people obtain the health services they need without exposing the users to financial hardship. There has been an increase in the number of programs and the way projects are designed, with a focus on equity aimed at improving access to and the utilization of maternal services among the poor and vulnerable in sub-Saharan Africa. However, the concept of equity and the way that poor and vulnerable are defined and applied in project implementation within specific contexts are not well known. The theory generated from our findings suggests that the impact of the PBF equity elements specific to maternal health depends on (i) a shared understanding of the definition of PAV among different stakeholders, including providers and users, as well as how the PAV is operationalized (structure), and (ii) the appropriate and timely payment of incentives to health facilities and health providers. The next step of this study is to assess the differential effect of PBF equity elements in accessing care using the proposed theoretical framework for data collection. 

## Figures and Tables

**Figure 1 ijerph-19-14132-f001:**
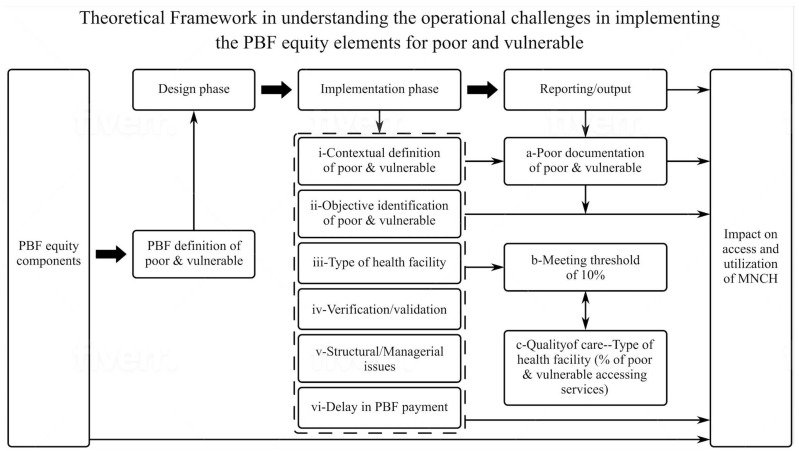
Theoretical framework in understanding the operational challenges in implementing PBF equity elements for PAV for maternal health.

**Table 1 ijerph-19-14132-t001:** PBF equity design elements and anticipated effect on equity.

PBF Equity Design Elements	Effect on Equity and Financial Protection
1. Choose services that are underused by the poor	Increased use of selected services by the poor
2. Pay providers more for reaching out to a poor person than a non-poor person	Increased use by the poor more than by the non-poor
3. Pay providers more for services delivered in poor areas	Increased use by people in poor areas more than by people in non-poor areas; more resources pushed to poor areas
4. Subsidize user fees	Reduce out-of-pocket cost, thus enhancing financial protection and increasing use of services
5. Incentivize community health workers	Overcoming information and social barriers for the poor
6. Add complementary demand-side incentives	Overcoming financial barriers (such as transportation cost and related expenses)

Source: Fritsche et al., 2014 [1].

**Table 2 ijerph-19-14132-t002:** Characteristics of health providers and administrators for key informant interviews. (a) Socio demographic characteristics of respondents for focus group discussion for women in Buea, Tiko, Limbe districts. (b) Socio demographic characteristics of respondents for focus group discussion for CHW in Buea, Tiko and Limbe districts.

		Tiko		Limbe		Buea	
Health care providers							
Nurses		2		1		2	
Midwifes		1		0		2	
Doctors		2		2		0	
Administrators including regulators and PBF managers		2		2		3	
Quarter heads		1		1		2	
PBF verificators		0		1		2	
PBF community verificators		1		0		1	
**Total**		**9**		**7**		**12**	
**(a)**							
**Participants**	**Age**	**Level of Education**	**Marital Status**	**Household Monthly Income (CFA francs)**	**Employment Status**	**Equity Status**	**District**
Code A	26	Primary	Single	<30,000	Not working	PAV	Buea
Code B	29	Primary	Married	<30,000	Not working	PAV	Buea
Code C	35	Secondary	Married	-	Housewife	PAV	Buea
Code D	35	Secondary	Married	-	Housewife	PAV	Buea
Code E	31	Secondary	Married	<30,000	Not working	PAV	Buea
Code F	39	Primary	Married	<30,000	Not working	PAV	Buea
Code G	38	Secondary	Married	-	Employed	NPAV	Buea
Code H	27	Secondary	Married	60,001–90,000	Employed	NPAV	Buea
Code I	26	University	Married	60,001–90,000	Employed	NPAV	Buea
Code J	28	University	Married	60,001–90,000	Employed	NPAV	Buea
Code K	33	University	Married	120,000–150,000	Housewife	NPAV	Buea
Participants	f						
Code A	33	Primary	Married	-	Not working	PAV	Tiko
Code B	29	Primary	Married	<30,000	-	PAV	Tiko
Code C	26	Secondary	Single	<30,000	Not working	PAV	Tiko
Code D	28	Secondary	Married	-	Not working	PAV	Tiko
Code E	21	Secondary	Single	<30,000	Not working	NPAV	Tiko
Code F	33	High School	Married	30,000–60,000	Not working	NPAV	Tiko
Code G	26	Secondary	Divorced	120,000–150,000	Self employed	NPAV	Tiko
Code H	26	High School	Married	120,000–150,000	Employed	NPAV	Tiko
Participants							
Code A	38	Primary	Married	<30,000	Not working	PAV	Limbe
Code B	37	Primary	Divorced	<30,000	Not working	PAV	Limbe
Code C	41	No education	Widowed	<30,000	Not working	PAV	Limbe
Code D	36	Secondary	Single	<30,000	Not working	PAV	Limbe
Code E	36	Secondary	Single	<30,000	Not working	NPAV	Limbe
Code F	38	Primary	Married	30,000–60,000	Self employed	NPAV	Limbe
Code G	41	Secondary	Married	-	-	NPAV	Limbe
Code H	39	Primary	Married	-	-	NPAV	Limbe
**(b)**															
**Participants**		**Age**	**Level of Education**	**Marital Status**	**Employment Status**		**District**
CHW1 Male		39	High School	Married	Employed		Buea
CHW2 Male		28	High School	Single	Employed		Buea
CHW3 Woman		38	-	Married	Not working		Buea
CHW4 Woman		39	Primary	Married	Employed		Buea
CHW5 Woman		27	-	-	Not working		Buea
CHW6 Woman		34	Secondary	Single	Employed		Buea
CHW7 Woman		32	Secondary	-	Not working		Buea
Participants							
CHW1Male		33	University	Married	Employed		Tiko
CHW2Male		41	High School	Married	Employed		Tiko
CHW3 Woman		42	High School	Married	Farming		Tiko
CHW4 Woman		43	Primary	Married	Employed		Tiko
CHW5 Woman		39	Secondary	-	Employed		Tiko
CHW6 Woman		33	Secondary	Married	Not working		Tiko
CHW7 Woman		29	Secondary	-	Employed		Tiko
CHW8 Woman		40	Secondary	Single	Not working		Tiko
CHW9 Woman		26	University	-	Student		Tiko
CHW10 Woman		28	Primary	Single	Not working		Tiko
Participants							
CHW1 Male		28	Secondary	Married	Employed		Limbe
CHW2 Male		43	Secondary	Single	Self employed		Limbe
CHW3 Male		49	Primary	Married	Employed		Limbe
CHW4 Woman		28	Secondary	Married	Not working		Limbe
CHW5 Woman		38	Secondary	Single	Employed		Limbe
CHW6 Woman		42	Secondary	Single	Farming		Limbe
CHW7 Woman		47	Secondary	Single	Farming		Limbe

Not working implies they do not have a contract with a health facility under the PBF program.

**Table 3 ijerph-19-14132-t003:** Themes generated from the data.

Themes	Categories	Examples of Codes
Implementation		
1. Contextual definition of PAV	Poor and vulnerable by facility	Poor, no job, orphanage, accident, internally displaced persons
2. Identification of PAV	Challenges in differentiating PAV, NPAV status by actors	Referral, home visits, PAV referrals from CHW not validated by health facilities, pretend to be PAV, quarter heads’ refusal to refer, quarter head’s, attitude with CHW, health administrator’s interrogation of PAV, poor documentation of PAV for validation
3. Type of health facility	Challenges private, public, confessional	Limited finance to support PAV, delay in payment, risk of no payment, influx of patient
4. Verification/Validation process	Challenges by CHW, health administrators, verifiers	Wrong referral, poor documentation, disagreement on definition of variables
5. Structural/Management	Experiences private facility, public, confessional	Referral for PAV, CHW, cost of services for PAV, delay in payment
6. Delay in payment	Equity bonus	Cannot support PAV, cannot pay staff
Other subcategories		
1. Poor documentation of PAV	Error margin, loss of incentive	Incomplete data, disagreement on definition of variables
2. Meeting threshold of 10%	Challenges private, public, confessional	Unpaid consultations, delay in payment, poor documentation
3. Quality of care	Private, public, confessional	Experiences with delay in payment, provision of care, volume of patients

**Table 4 ijerph-19-14132-t004:** Criteria for defining and identifying poor and vulnerable as reported by health facility administrators.

	Internally Displaced Persons	Physically Handicapped Persons	Inadequate Finance to Pay for Services at Point of Consultation	HIV Infected Persons (Especially Women)	Employment/ Occupation (Very Poor Background, Lack of Job)	Marital Status	Place of Residence	Social Network
Facility Type								
Public	×	×	×	×	×			×
Private for profit	×		×	×	×			×
Confessional	×	×	×	×	×	×	×	×
Para-Public	×		×	×	×	×	×	

Social network in this context referred mainly to family members, friends, and neighbors.

**Table 5 ijerph-19-14132-t005:** Responses from participants in defining and categorising poor and vulnerable.

Poor	Vulnerable
Income	Age
Household characteristics	Disability (physical)
Expenditures	Disease exposure/status, e.g., HIV
Social networks	Population group (internally displaced persons)
Occupation	Accident (for example, road accidents)
	Conflict setting

**Table 6 ijerph-19-14132-t006:** Using study findings to understand the PBF design elements and anticipated effect on equity for ANC and skilled birth delivery in Cameroon.

Design Elements	Effect on Equity and Financial Protection	Target/Potential Beneficiaries	Context of Implementation /Assessment	Level of Assessment Health Facility (Based on Study Findings)	Individual Level (Based on Study Findings)	Policy Implications from Findings
1. Choose services that are underused by the poor	Increased use of selected services by the poor.	Pregnant women, households, adolescent girls and women, health facilities, health providers	For ANC, skilled birth delivery, home visits, and community health worker referrals	Health facilities reported an observed influx of patients and potential increase in utilization in some health facilities but no evidence on the poor. However, facility reported provision of services to the poor which was not existing before PBF	Some women experienced increases in OOP expenses for first ANC and C-section after PBF due to the introduction of quality indicators. Women would love to reuse services for skilled birth delivery based on the quality and cost	Introduction of quality indicators for first ANC and skilled birth delivery is relevant for content quality but has a cost implication, which needs to be taken into consideration in a system with existing high OOP expenses [31]
2. Pay providers more for reaching a poor person than a non-poor person	Increased use by the poor more than the non-poor	Pregnant women, members of households, health providers	CHW referral, health facility. PBF pays 4× equity incentive for each indicator served to the poor	Objective identification of poor and vulnerable is required to ensure the 4× payment for indicators is targeted to the “poor”.	Criteria for identification of poor and vulnerable can be biased and may result in misclassification	Meeting the 10% threshold for some health facilities is a challenge with the delay in the payment of PBF incentives. Delay in payment may hinder health facilities to continually provide care to poor and vulnerable and impact quality.
3. Pay providers more for services delivered in poor areas	Increased use by people in poor areas more than by people in non-poor areas; more resources pushed to poor areas	Health areas, health facilities, pregnant women	The district/regional equity incentive is based on equity scores, which is assumed to have a trickle-down effect, as areas with high equity scores should receive more bonuses than those with low equity scores	Incentives provided to providers can stimulate demand, but delays in payment of the PBF incentives can potentially affect the provision of services to the poor and vulnerable	Change in cost and incentives provided can stimulate demand, but attitude of health providers (quality) is an important factor in determining utilization amongst the poor, especially their choice for skilled birth delivery [32]	Utilization of services by the poor is determined by not only a change in cost but also a function of quality and other factors including cultural [32]
4. Subsidize user fees	Reduce out-of-pocket cost, thus enhancing financial protection and increasing use of services	Health areas, health facilities, pregnant women	Cost for certain services such as ANC and skilled birth delivery is subsidized for the poor and vulnerable, and in some cases, the service is provided for free	Identification at health facility is subject to bias, no objective assessment to qualify patients. Mostly inexpensive items that have been subsidized for example, ion medication	Some women reported that they received free services and subsidized services, especially internally displaced persons	Some observed changes in cost for family planning and medication; however, the cost increased for first ANC and skilled birth delivery due to increase in the quantity of test to enhance quality [31]
5. Incentivize community health workers	Overcoming information and social barriers for the poor	Community health workers, pregnant women, health facilities, households	The use of community health workers, especially during home visits, facilitate identification of poor and vulnerable individuals, and women unable to seek care are referred to health facilities	Health providers are frustrated with CHW’s approach in the identification and referral process for poor and vulnerable	CHWs’ frustration in communication and understanding of the criteria for validating poor and vulnerable as provided by the health facility and the referral payment system	Providers assumed the need for a reinforced training on CHW on the equity package and clearly define criteria for identification of poor and vulnerable based on context. CHW proposed proper communication and documented procedures and appreciation of circumstantial situations
6. Add complimentary demand-side incentives	Overcoming financial barriers (such as transportation cost and related expenses	Health areas, health facilities, pregnant women,	One private facility reported this	Rare practice in all health facilities and almost non existing in the context	Internally displaced women reported receiving some baby support such as napkins after delivery. Institutional policy implemented due to PBF by the lone private facility to support PAV women	Demand-side financing is essential to balance supply-side strategy, especially in a context with high existing out-of-pocket expenses but important to consider universal and/or targeted approaches

Source: The first two columns are based on the PBF equity design elements from Fritsche et al., 2014 [1] (as shown in Table 1), and the remaining columns are based on research findings of the thesis, Examples are based on context from health facility in the study setting.

**Table 7 ijerph-19-14132-t007:** Proposed steps for data collection to assess effect of PBF on equity for maternal health.

Steps	Definition	Description
1. Definition of poor and vulnerable (contextual)	Consideration of how PAV is defined in the specific context within the PBF program and within the context in general	Poor is defined differently from one context to another; it is important to contextualize poor and vulnerable to guide data collection
2. Variable definition	Define the variables to be collected based on the characteristics in the definition of PAV, and consideration of the identification, and validation process of PAV	Consideration of factors described in the framework in Figure 1
3. Data collection stage 1	At health facility, identify those considered as PAV and non-PAV. Conduct exit survey using variables defined in step 2	Sample secondary data from health facility where PAV is validated to define the sample for PAV and non-PAV women
4. Data collection stage 2	Follow-up data collection at household for the same participants identified at health facility as PAV and non-PAV	Conduct household survey on the same participants for PAV and non-PAV to assess household characteristics and other relevant variables defined in step 2
5. Scale	Define the scale based on contextual definition of poor and vulnerable	Apply scale to assess differential effect on PAV and non-PAV
6. Analysis	Apply existing analysis approaches and quantify the effect on PAV and non-PAV using the defined scale	Compare differences and assess changes within and across groups using the contextual definition of PAV as defined in step 1

## Data Availability

Data relevant for the study are included and/or attached as Appendix A.

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
