# Peer review of "Examining the Implementation of the Performance-Based Financing Equity Strategy in Improving Access and Utilization of Maternal Health Services in Cameroon: A Qualitative Study"

_ijerph, 2022, doi:10.3390/ijerph192114132_

Round 1

Reviewer 1 Report

This paper is very interesting, presents the objectives very clearly, is very well structured and complete. The methods are presented in a very detailed, in-depth way, with the research options very well described and grounded, adding quality and credibility to the work.
The results are robust and bring new information to this area of research.
Despite the quality of the work and my clear recommendation for publication, in my view, the article could benefit and be improved by considering the following:
- in the framework, it could theoretically problematise the concepts of poverty and vulnerability from broader perspectives than those operationalised in the study.
- in the discussion, it would make sense to consider the importance of studying the impact of the PBF on the (perceived) health of individuals, of access to health care, of inequality in access to health.
- one detail: in line 415, after a quote, several question marks appear.

Author Response

Please see the attachment i

Reviewer 2 Report

Thank you for giving me the opportunity to read this interesting manuscript. The study aims to understand how equity — as predefined by the performance-based financing (PBF) programme — is understood, experienced and put into operation by local stakeholders and target populations in Cameroon to achieve the PBF programme goals. The secondary aims are to understand the potential impact of the programme and identify operational challenges. There are increasing number of health programmes and projects that aim to improve access and utilisation of healthcare services among the poor and vulnerable (PAV) in resource-poor setting. Yet, the concept of equity, as well as how PAV are defined and applied in project implementations are less known. The study aims to fill this gap which makes the manuscript highly relevant to research, practice and public health policy making.

Nevertheless, the manuscript in the current state has many shortcomings and major revision is required especially with regards to methods, data analysis and reporting of the results. I would recommend the authors to refer to the Domain 3 of the “Consolidated criteria for reporting qualitative research (COREQ): a 32-item checklist for interviews and focus groups”.

1. Typos, grammar and English expressions

I recommend the authors to carefully reread the manuscript and check typos, grammatical mistakes, delete or add spaces between words and check table numbers. Ex. Line 360 should be table 2, Line 396, “table xx”, etc… It is advised to ask an English native-speaker to proof-read the text. Below are some examples, just to point out (not an exhausted list).

  • Line 23 and 121: “defined” instead of “define”
  • Line 29: “Southwest or South West (SW)” instead of “SW”
  • Line 30, 32, 33, 161: It seems that spaces are missing between some words.
  • Line 51: “The intention is to facilitate the allocation of resources……” instead of “This implies…” – not clear what “this” is.
  • Line 56 and 69 start with identical sentences – reconsider.
  • Line 70: consider changing the word “bonus” to something clearer. Specify what “bonus” means: ex. extra payment made to facilities?
  • Line 72: I assume the authors mean: “to give the highest equity incentives”.
  • Line 124: “. [10].”
  • Line 289: “[21-25.”
  • Line 394: Not clear what Table 2 is.
  • Line 415: “??” is not appropriate.
  • Line 165: “HealthCARE is delivered BY both private….”
  • Line 538-539: Suggest another way of articulation (meaning unclear).

2. Clarity, relevance and structure

The manuscript is highly relevant for the field. Nevertheless, it lacks clarity due to too many dispersed research questions asked in one study, inappropriate application of the grounded theory approach and the insufficient way in presenting results and findings. The manuscript can be shortened as some parts are repetitive (for example, Line 141-146 is repeating line 131-136).

3. Citations and references

The cited references seem to include the most recent publications available and are relevant. Amount of self-citation is reasonable. Reference [10] (self-citation) makes up an important part of the literature review in the manuscript but is not a published study. Maybe it would make sense to cite the original articles.   

Line 455: Reference is missing.

4. Scientific soundness and design

Scientific soundness, especially with regards to the interpretative approach and the role of the researcher can be improved. The qualitative methodological design used to answer the study question is appropriate but the design does not follow best practice guidelines of qualitative study. The manuscript in the current state does not demonstrate authors´ full understanding to a grounded theory approach.    

5. Methods and reproducibility

As it is a qualitative study, study findings do not have to be reproducible but the procedures used for collecting data and analysing them need to be reproducible. With the current explanations given in the method section, it would be difficult to repeat the same procedure especially for the data analysis part.

Suggestions:

To claim as a grounded theory study, first, the author needs to clearly state that the aim is theory building: ex. developing theory on the process that influence the outcome/quality/impact of PBF interventions in the specific context of Cameroon and in the field of maternal health.

Second, the authors need to specify which of the three major schools of grounded theory approach has been used (1. Glaser, 2. Strauss and Corbin, or 3. Charmaz). As implied from one of the interview quotations, (line 498), the author/interviewer has been in direct contact with some of the service providers/interviewees and has been providing training on how to define PAV (this contradicts with the statement in line 273 that “no relationship was established between the interviewer and participant before the start of the interview”). An extensive literature review has been conducted prior to, or in parallel to data collection.

If this is the case, the Charmaz constructive approach will be the most appropriate. When this approach is chosen, the authors need to be very clear and open with their positions, prior knowledge, occupation, and relationships with the interviewees. This is missing in the method section.

Line 150: here the goal of theory building should be stated.

Line 152-158. How the PROGRESS perspective is compatible with the grounded theory approach is not clear. This paragraph is not about the study method but an explanation of the PROGRESS.

Sampling (line 171-192): This part can be substantially shortened. Instead of explaining that there are disagreements with regards to sample size, state whose definition was applied and why.

The authors claim that theoretical sampling was used but what kind of theory has been developed and applied in subsequent interviews are not made clear.

The authors collected data with multiple methods: individual interviews, focus group discussions, a PBF validation meeting and non-participant observation. It is not made clear how all these data were collated, in what order they were collected and how the preliminary developed theory was subsequently applied.

6. Figures/tables/images

I recommend all tables and figures to be reviewed and revised (together with corresponding parts in the text). For example:

  • Table 1. “choose” – changed to capital “C”. Perhaps indent to the left to make it easier to read). As mentioned before, table numbers in the texts do not match the actual tables presented.
  • Figure 1. Perhaps a simple table is sufficient and not a figure.
  • Figure 2. Recommend to make to simplify and make it more intuitively understandable.

7. Data analysis and interpretation

The authors claim to have followed a rigorous data analysis process with two coders and developing a codebook. Either the codebook can be provided as a supplementary material or examples of dominant/non-dominant codes as well as how these were developed into major/sub-themes should be given. There are repeated explanations that “constant comparison” was made but evidences are lacking how this was done (what was compared with what?). What codes/themes were applied to which quotes (provide some examples).

Line 312-327: How the codebook was developed and applied is unclear (codebook should not be the outcome of “exporting results of various themes”. Codes do not get exported as “codebooks” either – it is not an automatic process (More than one codebook was used?)

Figure 1. It is not made clear what kind of “themes” were developed from the codes. “Poor” or “Vulnerable” cannot be considered as themes from a qualitative analysis perspective.

The authors also collected data from a very different groups of stakeholders. How did this triangulation method contribute to theory building?

Results

Line 366: What are the “39 indicators”?

Perhaps it is clearer to report results per stakeholders: 28 providers of PBF, 27 pregnant women and 24 CHW.

Results could be presented per themes that leads to the theory. It is not clear what kind of “themes” were developed from the coding process (and the codes that make up these themes).

In the result section, the authors should clearly differentiate: 1) findings from the coding process, 2) authors´ interpretation of the data, 3) what the actors are expected to do (based on the PAB guideline) but what in fact they claim to do. In several points (ex. line 521) it is stated what the actors are “expected” to do, but this should not be presented as a “result” of the data analysis (or be made clear that this is part of the interpretation of the data).

Conclusions

The study implies that the following theory/hypothesis could be developed: the quality/impact of the PAB programmes with focus on maternal care depends, to a large extent, on two factors: 1) shared understanding to the definition of PAV (by both providers and users) as well as how the PAV is operationalised (operational structure) and 2) the appropriate and timely rewarding of providers (ex. regular payment).

These are valuable findings and a good theory. However, this is not made explicit, nor the current way of presenting the results do not sufficiently back-up this conclusion.

My suggestion would be to reorganise the whole manuscript that leads to this theory building following more rigorously Charmaz´s grounded theory approach.

Line 973-977: meaning is unclear. I recommend to stick to one theory as suggested above.

Ethics and data

Ethical statements and data availability statements seem adequate.       

Round 2

Reviewer 2 Report

Dear authors,

Thank you for sending the revised version and your response to my comments. The quality of the manuscript has improved, the tables are clearer, and I believe it makes an important contribution in the field of study. Below are further comments and suggestions.

Line 32 (Abstract) and Line 362-366: the authors can explain the number of participants to the study and how the data were collected in a clearer and simple way, as these are confusing (especially line 362-366 requires English changes). What is clear from the Tables do not have to be articulated in detail.

Suggestion: "Key informant interviews and focus groups were conducted with 79 participants, including 28 health professionals and service administrators, 27 pregnant women and 24 community health workers (CHW) in three districts"

Line 40, 408, 1042: "(both providers and usurers)" - consider elaborating further, such as "among different stakeholders, including providers and users". In the main text, maybe make it more explicit without using the brackets. 

Table 1. A space seems to be missing between "1." and "Choose"

Line 152-159: English suggestion (and shorter version) "This study employed Charmaz´s constructivist grounded theory approach which presumes researchers as co-producers of knowledge and not mere observers of the social processes. It assumes that theories and meanings derived from data are socially constructed, influenced also by the researchers´ interactions with the data as well as with their study participants. We yet remain with the principal that our interpretation remains grounded in the data taking fully into account of the context in which such data were collected [21, 23]."

Line 846-849: Please check the use of semicolon and English here.

Line 956-959: Please check the use of "--" and English here too. Line 957 should be under case "a" and not capital "A".

Line 1033-1043: Please check font (size).   

Line 1107: delete.
